# Virulent *Rhodococcus fascians* Produce Unique Methylated Cytokinins

**DOI:** 10.3390/plants8120582

**Published:** 2019-12-07

**Authors:** Paula Elizabeth Jameson

**Affiliations:** School of Biological Sciences, University of Canterbury, Christchurch 8140, New Zealand; paula.jameson@canterbury.ac.nz; Tel.: +64-275553582

**Keywords:** apical dominance, cytokinin, methylated cytokinin, *Rhodococcus fascians*, sugar will eventually be exported transporter, SWEET, amino acid transporter, sugar transporter, cell wall invertase

## Abstract

Some strains of *Rhodococcus fascians* exist only as epiphytes on the plant surface whereas others can become endophytic and cause various abnormalities including the release of multiple buds and reduced root growth. The abnormalities reflect the action of cytokinin. The strains that can become endophytic harbour a linear plasmid that carries cytokinin biosynthesis, activation and destruction genes. However, both epiphytic and endophytic forms can release cytokinin into culture, affect cytokinin metabolism within inoculated plants and enhance the expression of sugar and amino acid transporters and cell wall invertases, but only the endophytic form markedly affects the morphology of the plant. A unique methylated cytokinin, dimethylated *N*^6^-(∆^2^-isopentenyl)adenine (2-MeiP), operating in a high sugar environment, is the likely causative factor of the severe morphological abnormalities observed when plants are inoculated with *R. fascians* strains carrying the linear plasmid.

## 1. Introduction

Unique methylated cytokinins account for the morphological abnormalities induced by virulent strains of *Rhodococcus fascians*. For many decades, cytokinins produced by *R. fascians* have been implicated as the causative factors inducing shooty galls and reduced root growth, since application of cytokinin can mimic the disease symptoms [1]. Indeed, virulent strains harbour a linear plasmid that carries genes for cytokinin biosynthesis (*fasD*), cytokinin activation (*fasF*) and cytokinin destruction (*fasE*), and avirulent strains lack such a plasmid [2,3,4,5]. Cytokinin biosynthesis usually involves the attachment of an isoprenoid side chain to a molecule of either AMP (bacteria) or ADP/ATP (plants) by an isopentenyl transferase (IPT). Additionally, cytokinins are also found associated with specific tRNA molecules, synthesised via tRNA-IPTs [6,7,8]. 

Both avirulent (epiphytic) as well as virulent strains of *R. fascians* have been shown to extrude multiple different cytokinins into culture, most of which can, however, be derived from tRNA breakdown e.g., [9,10,11]. Likewise, multiple cytokinins can be extracted from plants inoculated with both virulent and avirulent strains e.g., [12,13,14,15]. Critically, the levels of individual cytokinins extracted from tissues inoculated with virulent strains have never been sufficiently elevated relative to mock-, or avirulent-inoculated plants to be convincing as the cause of the shooty galls. 

In a well-cited publication, it was suggested that virulent strains of *R. fascians* trick the plant into providing a compatible environment for the pathogen. The “Trick-with-the-Cytokinin-Mix” hypothesis is based on the accumulation of several cytokinins in tissue inoculated by a virulent strain. These cytokinins are more-or-less resistant to destruction by cytokinin oxidase/dehydrogenase (CKX) [16]. However, these data derive from a comparison between plants inoculated by a virulent strain and a mock-inoculated control, and lack a comparative analysis with avirulent-inoculated plants. 

We have shown that both virulent and avirulent strains of *R. fascians* can produce the cytokinins implicated in the ‘Trick-with-the-Cytokinin-Mix’ hypothesis *in planta* [13,14,15]. Moreover, both a virulent and an avirulent strain caused increased expression of amino acid (*AAP*) and sugar (*SWEET* and *SUT*) transporters and cell wall invertases (*CWINV*) [13,14]. A high sugar environment is required for release of apical dominance [17], so there is clearly something unique to the virulent strains that leads on to the initiation of the shooty galls and inhibition of root growth. 

In 2015, Sakakibara’s lab identified two previously unknown methylated cytokinins, monomethylated *N*^6^-(∆^2^-isopentenyl)adenine (1-MeiP) and dimethylated *N*^6^-(∆^2^-isopentenyl)adenine (2-MeiP), in tobacco tissues of plants inoculated with a virulent *R. fascians* strain [18]. They showed expression of two methyl transferases (*mt1, mt2*) and *fasD*, all from the linear plasmid of a virulent strain, were sufficient to produce 2-MeiP in transgenic *E. coli* cultures. FasD utilised the dimethylated side chain to produce 2-MeiP, indicating that FasD is a dimethyl transferase rather than an isopentenyl transferase [18]. 2-MeiP is resistant to degradation by CKX and inhibited root growth [18]. Moreover, Vereecke’s lab had earlier reported that *mt1* and *mt2* mutants of *R. fascians* were non-pathogenic [19]. We recently showed that both *mt1* and *mt2*, and *fasD*, are expressed in peas inoculated with a virulent strain but not in tissues inoculated with an avirulent strain or in controls. 1-MeiP and 2-MeiP were detected in pea tissues inoculated with the virulent strain [15].

We also showed that none of the cytokinins implicated in the ‘Trick-with-the-Cytokinin-Mix’ hypothesis correlated with virulence [15]. We suggested that 2-MeiP, a cytokinin produced uniquely by virulent *R. fascians* strains (ours and that used by Radhika et al., 2015) within the generally low level of cytokinins able to be produced by both virulent and avirulent strains, should be the subject of further analysis. As of now, assuming 2-MeiP is responsible for changes in the plant phenotype, it is the simpler explanation for the induction of the morphological abnormalities induced by virulent *R. fascians* and provides for a more readily testable hypothesis compared to the Trick-with-the-Cytokinin-Mix hypothesis. The interaction of 2-MeiP within a high sugar environment should be sufficient to initiate the substantive morphological differences seen between plants inoculated with virulent compared to avirulent strains of *R. fascians*, as depicted in Figure 1.

## Figures and Tables

**Figure 1 plants-08-00582-f001:**
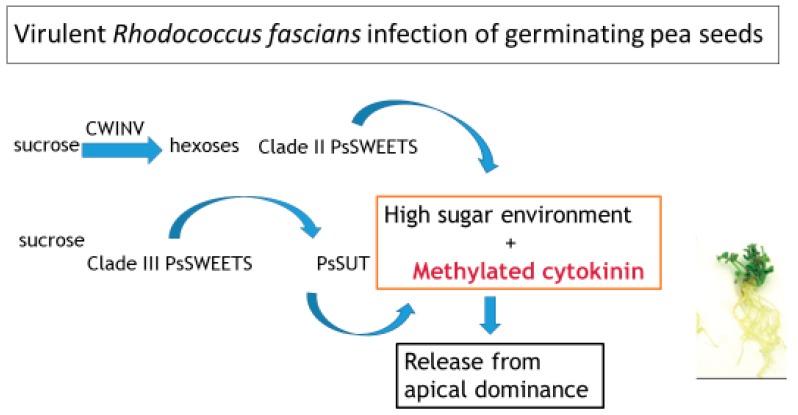
Model illustrating that the morphological abnormalities induced by virulent *Rhodococcus fascians* strains are caused by the production of a novel methylated cytokinin by *R. fascians* in a high sugar environment resulting in the release of apical dominance.

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
