# Peer review of "Virulent *Rhodococcus fascians* Produce Unique Methylated Cytokinins"

_plants, 2019, doi:10.3390/plants8120582_

Round 1
Reviewer 1 Report
The manuscript deals with a relevant subject to PLANTS. The ms is very interesting and clear, well written, and the appropriate topics are supported by the literature. I recommend that the manuscript should be accepted.
Minor points:
Line 17: authors should include in full the meaning of 2-MeiP.
Line 111: authors should remove 2010.
Author Response
The full meaning of 2-MeiP is now provided on line 17 of the Abstract; additionally the full meanings of both and of 1-MeiP and 2-MeiP are now provided in the text.
Line 111: 2010 has been removed.
Reviewer 2 Report
The commentary „Virulent Rhodococcus fascians produce unique methylated cytokinins” by Paula Elizabeth Jameson discusses the involvement of cytokinins produced by Rhodococcus fascians in induction of the morphological abnormalities in the plant host. The Author, basing on the recently published works, proposes a novel hypothesis stating that the methylated cytokinin, 2-MeiP, is responsible for changes in the plant phenotype. The proposed model of induction of the morphological abnormalities is interesting and likely. The commentary is well written, concerns an important issue and should be of interest to a wide range of readers, as well as stimulating to further research.
Author Response
No changes requested.